Local network connectivity optimization: an evaluation of heuristics applied to complex spatial networks, a transportation case study, and a spatial social network

Auerbach Jeremy jeremy.auerbach@colostate.edu 1
Kim Hyun 2
1 Department of Environmental & Radiological Health Sciences, Colorado State University , Fort Collins , CO , United States of America
2 Department of Geography, University of Tennessee , Knoxville , TN , United States of America
Xia Feng
Electronic publication date: 2021 Jun 18
Publication date: 2021
Volume: 7
Electronic Location ID: e605
Received 2020 Jul 28; Accepted 2021 May 31
Copyright: ©2021 Auerbach and Kim
Copyright year: 2021
Copyright holder: Auerbach and Kim
License: This is an open access article distributed under the terms of the Creative Commons Attribution License, which permits unrestricted use, distribution, reproduction and adaptation in any medium and for any purpose provided that it is properly attributed. For attribution, the original author(s), title, publication source (PeerJ Computer Science) and either DOI or URL of the article must be cited.
License URL: https://creativecommons.org/licenses/by/4.0/

Keywords: Networks, Network connectivity, Transportation, Urban planning, Genetic algorithms, Street connectivity, Transportation networks, Network optimization, Social networks, Spatial networks

Funding: The authors received no funding for this work.

==============================
Optimizing global connectivity in spatial networks, either through rewiring or adding edges, can increase the flow of information and increase the resilience of the network to failures. Yet, rewiring is not feasible for systems with fixed edges and optimizing global connectivity may not result in optimal local connectivity in systems where that is wanted. We describe the local network connectivity optimization problem, where costly edges are added to a systems with an established and fixed edge network to increase connectivity to a specific location, such as in transportation and telecommunication systems. Solutions to this problem maximize the number of nodes within a given distance to a focal node in the network while they minimize the number and length of additional connections. We compare several heuristics applied to random networks, including two novel planar random networks that are useful for spatial network simulation research, a real-world transportation case study, and a set of real-world social network data. Across network types, significant variation between nodal characteristics and the optimal connections was observed. The characteristics along with the computational costs of the search for optimal solutions highlights the need of prescribing effective heuristics. We offer a novel formulation of the genetic algorithm, which outperforms existing techniques. We describe how this heuristic can be applied to other combinatorial and dynamic problems.

Introduction

Spatial networks have become more popular as the interest in networks has spread into more fields and spatial data, and the computational power and methods to analyze it, have become more accessible. In terms of analysis, spatial network optimization has been at the forefront and focused on increasing network connectivity and information flow (Schrijver, 2002; Wu et al., 2004). Heuristics have been developed to rearrange existing networks or creating new ones that optimize the topology of the network for synchronizability (Khafa & Jalili, 2019). Several effective methods have also been developed to add new edges to a network that minimize the average shortest path distance (Meyerson & Tagiku, 2009), minimize the network diameter (Demaine & Zadimoghaddam, 2010), or maximize the network’s centrality (Jiang, Liang & Guo, 2011) or connectivity (Alenazi, Çetinkaya & Sterbenz, 2014).

While this optimization of spatial networks’ global characteristics, optimizing local existing network connectivity around a specific node or location with the introduction of costly new edges has not been explored and yet is important in several domains. For example, increasing an existing network’s connectivity around a focal node while minimizing the costs associated with the number and lengths of additional connections is essential in network layout planning for telecommunications and computer systems (Resende & Pardalos, 2006; Donoso & Fabregat, 2007) and increasing or slowing the spread of information or diseases in social networks (Gavrilets, Auerbach & van Vugt, 2016; Eubank et al., 2004). This local network connectivity problem is particularly important with transportation planning in urban environments, where the weights of the network edges can be physical distances or riderships and future street connections or transportation lines can be significantly costly and impact flow to established facilities. For example, planners can optimize thoroughfare connectivity around schools to foster student walking and biking while reducing busing costs (Auerbach, Fitzhugh & Zaviska, 2021) and increase accessibility and patient travel time to health care facilities (Branas et al., 2005).

The search for new edges that maximize connectivity to a focal node and minimize the costs of these new edges is not well understood and this search for optimal solutions can become costly when networks are large and complex. To fill this knowledge gap, we compare a set of heuristics to optimize local network connectivity applied to real-world networks and randomly generated ones. These heuristics are drawn from Mladenović et al. (2007) review of combinatorial heuristics and from location models that include a spatial component (Brimberg & Hodgson, 2011). We also offer a genetic algorithm with a novel chromosome formulation where the genes are not properties of a specific variable but weights for the probability to move in a given dimension across the solution space.

These optimization heuristics are then applied to randomly generated networks that vary in complexity and size to evaluate their efficacy in finding the optimal new connections that maximize local connectivity. Included in this set of random graphs we provide two novel formulations of random planar networks based on the Voronoi diagram and the Delaunay triangulation. To complement the random network analysis, the network connectivity optimization methods are also applied to two real-world case studies, one from urban transportation planning and another from social network analysis. In this study, we show that optimization heuristics are preferred for the analysis and practice due to the nonlinearity of the solution space and the optimal solution’s dependence on nodal characteristics, such as distance to the focal node. The novel genetic algorithm outperformed the other heuristics as it was able to move from suboptimal solutions and explore distance solutions quicker. This is important as researchers and engineers are working with networks or growing complexity and size.

The organization of this paper is as follows. The next section describes the formulation of the connectivity problem in more detail, the local search methodology, and the optimization heuristics (see Appendix A in the Supplemental Information for the specific pseudocode of the optimization algorithms). This is followed by a section that details the data used for the study including descriptions of the random networks, the transportation case study street networks, and the social network data. Results of the heuristics applied to the random networks and the case studies are then presented. The paper concludes with a detailed discussion of these heuristic results, the further implications of these techniques for urban transportation planning, and future work for this avenue of research.

Methods and Data

Formulation of the network connectivity problem

For the description of the optimization methodology the following nomenclature will be used (see Table 1). In connectivity optimization, network nodes are first segmented and assigned to ’close’ and ’distant’ sets by a chosen threshold distance D from the network’s focal node F. The number of nodes ν is a network is N, and nodes are separated into two sets based on their shortest network path distances to the focal node, d(ν, F). The nodes that are within this distance are assigned to the ‘close’ set, NC ⊂ N, i.e., ν ∈ NC if d(ν, F) ≤ D and F ∈ NC (see Fig. 1A). The nodes that are outside the threshold shortest network path distance to the focal node, D, are assigned to the ‘distant’ set, ND ⊂ N, i.e., ν ∈ ND if d(ν, F) > D. When a new connection is added to the network, the shortest path distance from each distant node to the focal node is recalculated. If there are any distant nodes that are now within the threshold distance to the focal node they are assigned to the new set Ni,jC. For example, if a new connection is established between distant node i and close node j, then k∈Ni,jC if k ∈ ND and d(k, i) + d(i, j) + d(j, F) ≤ D (Fig. 1C). The benefit of this new connection is B(i, j) and the cost associated with the new connection is C(i, j). The optimal solution is the solution with the greatest benefit, or number of new nodes now within the distance to the focal node which can be expressed as the bi-objective function (1) O∗=maxBi,j+ minCi,js.t.∑iνi=1,i∈NC ∑jνj=1,j∈ND

where benefits dominate costs. For example, if B(i, j) = B(m, n) and C(i, j) < C(m, n), then the optimal additional edge is between i and j. For nondominated solutions, we select the solution that minimizes the distance to the so-called ideal point. The ideal point represents the solution that simultaneously maximizes the benefit and minimizes the cost. For the analysis in this paper the formulation of the objective function is as follows. For a new edge between i and j the number of nodes in Ni,jC set is the benefit of this new connection, Bi,j=|Ni,jC|, and the cost of the new connection is the length of the edge C(i, j) = d(i, j). Therefore, (2) O∗=max|Ni,jC|+ mindi,js.t.∑iνi=1,i∈NC ∑jνj=1,j∈ND

Most of the heuristics presented below are dependent on the number of iterations (t) and terminate when the solutions converge, Ot = Ot−1, or the solution does not improve, Ot < Ot−1.

Local search methodology

For a network of size N the number of new connections to evaluate has an upper bound of N2∕4, therefore heuristics may be employed to identify (nearly) optimal solutions quicker than an exhaustive search as networks get larger. These optimization algorithms require a search space to explore and using nodal characteristics we create such a multidimensional solution space (see Table 2). These nodal characteristics are explored to find the critical network properties for connectivity optimization and their impact on the performance in finding the optimal solution. As several nodes in a network can potentially have the same nodal characteristic values the local searches include a random shuffling routine. In detail, the following individual node characteristics were used to create the dimensions of the solution space: (i) distance to the focal node, (ii) degree centrality, (iii) closeness centrality, (iv) betweenness centrality, (v) eigenvector centrality, (vi) pagerank centrality, (vii) weighted clustering coefficient, and(viii) the neighbor nodes directly connected to the node. Nodes were sorted by their distance to the focal node and moving in this solution dimension may result in lower connectivity length costs but may not maximize the number of nodes ultimately connected to the focal node. In contrast, sorting nodes by their centrality, i.e., the importance of the node, could result in maximizing the number of nodes within the specified distance to the focal node but with the possibility of higher connectivity length costs compared to selecting nodes by other characteristics. To avoid these two extreme possibilities, several commonly used measures of centrality are explored: degree, the number of edges incident to a node; closeness centrality, the average length of the shortest path between the node and all other nodes in the network (Bavelas, 1950); betweenness centrality, the frequency of a node included in the shortest paths between all other node pairs (Freeman, 1977); eigenvector centrality, which is a relative sorting of nodes such that nodes with high values are connected to other nodes with high values (Newman, 2008); and pagerank centrality, a variant of eigenvector centrality that sorts nodes based on their probability of being connected to a randomly selected node and which is commonly used in web-page rankings (Brin & Page, 1998). The weighted clustering coefficient of a node is the count of the triplets in the neighborhood of the node and accounts for the weights of the edges times the maximum possible number of triplets that could occur (Barrat et al., 2004). The weights used here are the spatial distances between nodes wij = d(i, j). The nodes that were directly connected to the node under evaluation were also used.

Table 1 List of symbols and their definitions.

Symbol	Definition	
ν	Network node	
e	Network edge	
N	Number of nodes in a given network, N = ∑iνi	
A	Network adjacency matrix	
aij	Adjacency matrix element ij	
F	Focal node	
d(i, j)	Network distance between nodes i and j	
D	Threshold distance from focal node	
NC	Set of close nodes, NC ⊂ N	
ND	Set of distant nodes, ND ⊂ N	
Ni,jC	Set of nodes that are now close after a new connection between nodes i and j	
LF	Average path length to the focal node	
C(i, j)	Cost of the new connection	
B(i, j)	Benefit of the new connection	
α	Cost weight	
β	Benefit weight	
t	Optimization iteration	
Ot	Optimal solution for iteration t	
O∗	Optimal solution	
M	Set of long-term memory solutions	
CiD	Degree centrality of node i	
CiC	Closeness centrality of node i	
σij	Shortest path between nodes i and j	
σjk(i)	Shortest path between nodes j and k that includes node i	
CiB	Betweenness centrality of node i	
λ	Eigenvalue	
xi	Eigenvector	
CiE	Eigenvector centrality of node i	
αP	Attenuation factor	
CiP	Pagerank centrality of node i	
η	Variable neighborhood size	
μ	Genetic algorithm mutation rate	
s	Genetic algorithm selection coefficient	
P	Population of solutions for the genetic algorithm	
f(i, j)	Genetic algorithm fitness function	
ϵB	Benefit error from heuristic	
ϵC	Cost deviation from heuristic	
p	Connection probability (Erdös-Rényi graphs and Klemm and Eguílez graphs)	
pW	Rewiring probability (Watts-Strogatz graphs)	
kL	Initial node degree (Watts-Strogatz graphs)	
m0	Initial network size (Barabási and Albert graphs and Klemm and Eguílez graphs)	
m	Degree of new nodes (Barabási and Albert graphs)	
pS	node selection probability (Klemm and Eguílez graphs)	
pR	Edge removal probability (Delaunay and Voronoi random graphs)	
CD¯	Mean degree of a network	
L¯	Average path length of a network	
ciw	Weighted clustering coefficient for node i	
wij	Weight of connection between nodes i and j	
C¯	Weighted clustering coefficient of a network	
Cr¯	Weighted clustering coefficient of a completely random network	
Lr¯	Average path length of a completely random network	
γ	Power law exponent	
P(n)	Degree distribution	
E	Efficiency of a network	
Er	Efficiency of a completely random network	
EG	Global efficiency of a network	

Network connectivity optimization heuristics

The following techniques were implemented for the network connectivity optimization study from their extensive use in optimization: hill climbing with random restart (Russell & Norvig, 2004); stochastic hill climbing (Greiner, 1992); hill climbing with a variable neighborhood search (Mladenović & Hansen, 1997); simulated annealing, which has a history of applications in graph problems (Kirkpatrick, Gelatt & Vecchi, 1983; Johnson et al., 1989; Kirkpatrick, 1984); and genetic algorithms, which has been successfully used for combinatorial optimization (Anderson & Ferris, 1994; Jaramillo, Bhadury & Batta, 2002). A Tabu heuristic was not employed as it has been observed to be an inferior method for multi-objective optimization problems compared to simulated annealing and genetic algorithms (Golden & Skiscim, 1986; Kim et al., 2016). Parameter selection was simplified for easy comparison of the methods (see the Supplemental Information for the algorithms). To ensure that the heuristics did not converge on suboptimal solutions due to the initial starting values, random restart, i.e., randomly selecting initial nodes to avoid local optima and running the routine until the optimal solution is found, was used.

Figure 1 Diagram of the sequence of the network connectivity optimization problem.

The close nodes that are within a threshold network distance (orange dashed circle) from the focal node (black square) are colored green, distant nodes that could be within the threshold network distance with additional edges are colored red, and the gray distant nodes are outside the threshold distance regardless of any additional connections. (A) is an example graph, (B) shows the same graph with the optimal new connection that maximizes the number of additional nodes within the threshold network distance and minimizes the length of the new connection, and the inset (C) highlights this optimal connection, between nodes i and j.

Table 2 Node characteristics used for the neighborhood search and their formulations.

Local search selection criteria	
distance from focal node		degree centrality	
d(i, F)		CiD=∑jAi,j	
closeness centrality		betweenness centrality	
CiC=1∕∑jdi,j		CiB=∑j≠i≠kσjkiσjk	
eigenvector centrality		pagerank centrality	
CiE=1λ∑jAi,jxj		CiP=αP∑jAi,jxj∑iAi,j+1−αPN	
weighted clustering coefficient	
ciw=1CiD−1∑jaijwij∑j,hwij+wih2aijaihajh	

Exhaustive search (ES). The exhaustive search calculates the solution for every pair of distant and close nodes for a network (see Algorithm 1 in the Supplemental Information). While this approach ensures that the optimal edges are found, as the number of nodes increases and therefore the number of possible connections between close and distant nodes increases, it can become computationally expensive and timely to implement. Since the results of ES provide the optimal solution, the times it takes to find evaluate all solutions are used to benchmark the other heuristics.

Hill climbing (HC). The solution space was observed to be hilly from the exhaustive search results, so several modifications were introduced to the hill climbing technique to avoid getting stuck in suboptimal solutions (Algorithm 2 in the Supplemental Information). A stochastic hill climbing (HCS), an advanced search method based on HC, routine is also explored where the selection of nodes for the next iteration is randomly picked with (3) probabilityi,j=Oi,j∑m,nOm,n,

which terminates when an improved solution is no longer found (Algorithm 3 in the Supplemental Information). A hill climbing algorithm is coupled with a variable neighborhood (HCVN) where the size of the neighborhood starts with the nearest neighbors (η = 1) and is updated as follows: (4) η=1ifOt>Ot−1η+1ifOt≤Ot−1,

and the HCVN method terminates after nmax is reached (Algorithm 4 in the Supplemental Information).

Simulated annealing (SA). As a meta-heuristic approach, the simulated annealing method randomly selects an initial solution from the solution space to avoid entrapment in a local optima. At each iteration, the heuristic evaluates the neighboring solutions and if it does not find an improved solution, it moves to a new solution with the following probability: (5) probabilityi,j= exp−Ot−1−Oi,jt.

The distance of the move decreases with the number of iterations until a better solution is no longer found (Algorithm 5 in the Supplemental Information).

Genetic algorithm (GA). The genetic algorithm begins with a population of P randomly selected solutions with a set of chromosomes composed of genes which represent the weights of selecting a neighbor and are all initialized to unity (Algorithm 6 in the Supplemental Information). During each iteration of the method, solution scores (fitnesses) are computed by (6) fi,j=Oi,j∑m,nOm,n,

and a new generation of solutions are selected based on the following probability condition (7) probabilityi,j=s∗fi,j+1−s∑m,ns∗fm,n+1−s,

where s is the selection coefficient. Weak selection, s ≪ 1, is used to ensure that random mutations impact solution frequency. Crossover is conducted by alternating the weights for the offspring from each parent, also known as cycle crossover (Oliver, Smith & Holland, 1987). Mutations are introduced at a low rate μ ≪ 1 for each gene and increase the nodal characteristic selection weight by one. The probability that characteristic m is used to find a neighbor for node i is given by (8) probability(characteristic)=genei,m∑kgenei,k∕K,

where K is the total number of nodal characteristics. This formulation ensures that the nodal characteristics that improve the solution increase in weight which results in a greater probability they will be selected for neighborhood exploration, and ultimately reduces the size of the neighborhood search. Among these methods, the genetic algorithm presented here introduces a novel chromosome formulation where the genes are not properties of a specific variable but weights for the probability to move in a given direction in the solution space. This allows the method to occasionally explore different nodal characteristics (at mutation rate μ) while conducting a local neighborhood search.

Simulated data: random networks

To test the efficacy of these optimization heuristics in finding the optimal new network connections they were applied to randomly generated networks that vary in complexity and size. Several types of random graph networks were generated to analyze the efficacy of the optimization heuristics for systems with different topologies which are generally representative of naturally occurring and built systems: (1) Erdös-Rényi networks, (2) Watts-Strogatz networks, (3) Barabási and Albert networks, (4) Klemm and Eguílez networks, (5) Delaunay triangulation networks, and (6) Voronoi diagrams. Erdös-Rényi random networks are constructed by randomly creating connection between pairs of nodes with a probability (Erdös & Rényi, 1959). These networks, even though they have random connections, consistently have short average path lengths and irregular connections, both of which are well found in natural systems. The Watts-Strogatz networks also have random connections but the networks also form clusters, another feature commonly found in real-world networks (Watts & Strogatz, 1998). The Barabási-Albert model produces random structures with a small number of highly connected nodes, ’hubs’, which are observed in numerous types of networks (Barabási & Albert, 1999; Albert & Barabási, 2002). Klemm and Eguílez networks have random connections, clusters, and hubs (Klemm & Eguílez, 2002). See Figs. S.1A–S.1D and Prettejohn, Berryman & McDonnell (2011) for the algorithms used to generate these networks.

We also introduce two novel types of random planar network versions of the Voronoi diagram and the Delaunay triangulation (Figs. S.1E and S.1F). Planarity is particularly important in many fields and networks generated from Voronoi diagrams and Delaunay triangles have been used in spatial health epidemiology (Johnson, 2007), transportation flow problems (Steffen & Seyfried, 2010; Pablo-Martì & Sánchez, 2017), terrain surface modeling (Floriani, Falcidieno & Pienovi, 1985), telecommunications (Meguerdichian et al., 2001), computer networks design (Liebeherr & Nahas, 2001), and hazard avoidance systems in autonomous vehicles (Anderson, Karumanchi & Iagnemma, 2012). Delaunay triangulation maximizes the minimum angles between three nodes to generate planar graphs with consistent network characteristics while Voronoi diagrams, the dual of a Delaunay triangulation, are composed of points and cells such that each cell is closer to its point than any other point (Delaunay, 1934). To modify these edges are removed from network nodes randomly based on their distance from the focal node with probability (9) pR⋅maxdi,F,dj,Fmaxkdk,F,

where pR is the removal probability and weighted by the normalized edge distance from the focal node. When edges are randomly removed from the connected Delaunay network or Voronoi network, with weights given by node distance from a focal node, these networks display some of the properties similarly found in the networks mentioned above, such as complexity and randomness. Yet, these networks have the added component of being planar and having edge weights that can be framed as physical distances.

To compare the efficacy of different optimization methods for different network topologies, identifying the best set of parameters are critical. Parameter values were selected for each type of random network to ensure network complexity (Table S.6 summarizes the parameters which were used in the analysis). Variation in network size was also explored and the most connected node in each network was selected as the focal node. Uniformly randomly generated edge weights in [0,1] were used for the network distances and the threshold distance was set to ensure that half of the nodes were initially within the distance to the focal node. The costs and benefits were normalized using the ranges from the exhaustive search routine as a benchmark to compare the results from the different optimization methods.

Empirical data: transportation case study

To complement the random network analysis, the network connectivity optimization methods were applied to a study of urban transportation planning. Network connectivity optimization methods were used to evaluate the potential costs and benefits of increased thoroughfare connectivity for student walking or biking to school. It is assumed that expanding the connectivity around a school would allow for more households, and students, to be included within the walking distance to the school. If more students actively commute to school, this reduces the busing costs for the school system and increases the health and academic achievement of the students (Centers for Disease Control and Prevention, 2010). Yet, streets have associated land, construction, and maintenance costs that are primarily dependent on their length.

Networks composed of street edges and residence nodes around several schools from a representative US school system were used for the analysis. Ten suburban and rural schools from Knox County, TN, were selected for the analysis, including seven elementary and three middle, that would benefit the most from increased thoroughfare connectivity, i.e., had the most students within the Euclidean walking distance but not the network distance to the school (characteristics of these schools are provided in the Table S.3). Urban schools were not used since the street connectivity around the schools was significantly high and the from additional thoroughfares would be low. Residential parcels and street networks were provided by the Knoxville-Knox County Metropolitan Planning Commission and the residential parcels were converted to nodes and placed on the nearest street edge. The residences within 1 mile and 1.5 miles, for the elementary schools and the middle schools respectively, are considered close nodes while the nodes outside of these distances were classified as distant nodes (see Fig. 2). The school networks do not generally display the characteristics of complex network: they had low average degree, large path lengths, and were not efficient, yet they did have power distributions of connectivity with few intersections having a large number of street connections (see Supplemental Information Table S.3). The networks were evaluated with each optimization method to maximize the number or close residences connected to the school and minimize the distance of the new thoroughfare. The costs and benefits of these street connections were normalized using the ranges from the exhaustive search routine as a benchmark to compare the results from the different optimization methods.

Figure 2 Results of the transportation case study used for the analysis.

A network of streets and residences around a school is shown in (A) and with the optimal new walking connection in (B). The red nodes represent the distant residences, i.e., the residences within the 1-mile Euclidean walking distance to the school but not the 1-mile street network walking distance, the green nodes are the close residences within the street network school walking distance, and the black square represents the school. The orange line is the optimal new walking connection that maximizes the number of additional residences (orange nodes) and minimizes the length of the new connection.

Empirical data: social network analysis

The topology of a network and the management of its system can improve information flow and have been shown to help counter the negative effects of social and environmental crises (Helbing et al., 2015). Specifically, increased social network connectivity can reduce the time information spreads among its members. For example, quickly notifying those nearby and unaware of an active shooter event can save lives. Using a set of social network data coupled with location data, we evaluated the heuristics to optimize social network connectivity for those nearby a specific location.

The data set used for this analysis was from Gowalla, a location-based social networking website where users share their locations by checking-in, and this data set was collected and published by Cho, Myers & Leskovec (2011). This social network is undirected and consists of 196,591 nodes (members) and 950,327 edges (social connections). There is a total of 6,442,890 check-ins of these members over the period of February 2009 to October 2010 (network characteristics are provided in the Supplemental Information Table S.4). For the analysis we simulated 100 crisis incidents at a highly populated urban place, Grand Central Terminal in New York City (NY). Grand Central Terminal was selected as the location of the simulated events as it is a major transportation hub located in the center of the city that serves over a million commuters and visitors daily. New York City is also a major metropolis with tens of thousands of Gowalla users present in the data set and the city has a history of incidents, such as terror attacks. Ten dates were selected at random and for each date ten times were randomly selected between 1200 and 1800 local to simulate a crisis event (see Fig. 3).

Figure 3 Results of the Gowalla social network case study used for the analysis.

For simplicity the network shown is the users with a check-in within 1 hour prior to the event and nearby (within 0.5-mile) the event location, Grand Central Station (NYC). (A) The network prior during the event and (B) with the optimal connection. The red nodes represent the users within 0.5-mile who are unaware of the event, the green nodes are the users aware of the event, the black line represents. The orange line is the optimal new social network connection that maximizes the number of additional nearby users aware of the event.

The social network problem was formulated such that an event occurred at the location (Grand Central Station) and members who were within 0.5-miles of the event should be notified. The members within Grand Central Station are automatically notified of the event whereas those outside can be notified through their online social network if a member in their network is aware. The event is considered to be serious enough that those members aware of it will share the information on the social network. New connections are evaluated based on their number of additional nearby members who become aware of the event.

Performance of the algorithms

Several finding are worthy to note regarding the performance of heuristic algorithms used in the analysis. First, there were consistent nonlinear relationships between the nodal characteristics and the quality of the solutions for each type of random network and the school networks (see Fig. 4). There was also significant variation for which nodal characteristics were correlated with the quality of the solution across networks (see Table 3). Among those, the distance between the close node and the focal node and the distance between the distant node and the close node were most often highly correlated with the quality of the solution across networks. The centrality measures were inconsistently related to the solution quality for the random networks yet were related to the optimal solutions for the social networks.

Figure 4 The relationship between the distances of the close node and the focal node with the costs and benefits for each solution for different networks.

(A) shows the relationship for a Watts-Strogatz network with N = 500, (B) a Barabási and Albert network with N = 500, (C) a Delaunay network with N = 500, and (D) a suburban school transportation network (N ≈ 4000). Each point represents a connection between a distant and close node, where the cost is the standardized length of the connection and the benefit is the number of new nodes within the distance to the focal node or school.

Table 3 Mean correlation coefficients for the nodal characteristics and the solution benefits for the experimental networks.

The three coefficients with the largest magnitude are highlighted in bold for each network type.

			Network	
			ER	WS	BA	KS	DT	VD	Schools	Social Network	
Nodal characteristics	Close node	d(j, F)	−0.08	−0.52	−0.37	−0.48	−0.28	−0.41	−0.30	0.11	
CjD	0.14	0.08	0.05	0.09	−0.01	0.02	0.02	0.09	
CjC	0.10	0.26	0.05	0.15	0.15	0.15	0.18	0.09	
CjB	0.14	0.15	0.05	0.04	−0.02	0.03	0.06	0.05	
CjE	0.12	0.11	0.05	0.09	0.01	0.08	−0.01	−0.07	
CjP	0.14	0.07	0.05	0.10	−0.03	−0.01	−0.00	−0.08	
cjw	0.00	−0.03	0.02	0.00	0.14	0.03	*	−0.04	
Distant node	d(i, F)	−0.46	−0.18	−0.12	−0.09	−0.07	−0.08	−0.04	−0.07	
CiD	0.17	0.03	0.08	0.05	0.29	0.30	0.08	−0.30	
CiC	0.15	−0.01	0.07	0.00	0.35	0.21	0.06	0.26	
CiB	0.19	0.03	0.07	0.03	0.20	0.20	0.07	−0.09	
CiE	0.21	0.02	0.07	0.06	−0.16	−0.01	−0.03	−0.10	
CiP	0.17	0.01	0.06	0.07	0.30	0.22	0.01	0.05	
ciw	0.08	−0.01	0.03	0.03	0.12	0.12	*	−0.08	
Notes.

* There was no variation in clustering coefficients as triplets were not common in the street networks.

The results of the termination times and the optimal solutions deviations from the optimization heuristics applied to the random networks are summarized in Fig. 5 and Figs. S.1 and S.2. The hill climbing method was consistently faster for all of the networks, yet had the largest cost and benefit deviations. Simulated annealing and the genetic algorithm had similar termination times, but the genetic algorithm was consistently superior to all of the other methods in approaching the optimal solution. The results from the application of the optimization heuristics applied to the ten school networks are shown in Figs. 5E and 5F. The times to termination for each heuristic according to network size consistently followed the following pattern: ES >SA >HCVN >GA >HCS >HC. The genetic algorithm clearly outperformed the other heuristics, followed by simulated annealing, in terms of cost and benefit deviations (see Figs. 5B, 5D, and 5F).

Figure 5 The termination times (A, C, and E) and the differences between the heuristic solution and the global optimal solution costs and benefits (B, D, and F) for the heuristics applied to a sample of random networks and school networks: (A) and (B) Erdös-Rényi networks, (C) and (D) Delaunay networks, and (E) and (F) the ten school networks.

For the random networks (A and C) the termination times were averaged over 1,000 networks for network sizes of 500, 1,000, and 2,000 nodes. The times were normalized by the exhaustive search time and log transformed. For the random network cost and benefit differences (B and D) were drawn from a sample of 100 networks with 1000 nodes. The costs were the total lengths of the new connections and the benefits were the number of additional nodes for the solution. The costs and benefits were normalized with the global optimal solution from the exhaustive search, and a longer connection length is a positive cost difference whereas a shorter connection is a negative cost difference. For visualization, the 95-percent confidence ellipses drawn from the Hotelling’s T 2 statistic were included. The heuristic acronyms are the following: exhaustive search (ES), hill climbing (HS), stochastic hill climbing (HCS), hill climbing with variable neighborhood (HCVN), simulated annealing (SA), and genetic algorithm (GA).

Discussion, future work, and limitations

The local network connectivity problem introduced in this study is relevant to a wide range of applications and is nontrivial as the number of potential solutions can become large even for small systems. This class of combinatorial optimization problem highlights the difficulty in determining local search routines a priori. When the exhaustive search routine was applied to random networks and the real-world networks, the optimal solutions were found to be related to nodal characteristics, which entails a great complexity to find optimal solutions. Therefore, the heuristics employed to reduce the computational costs utilized nodal characteristics to search for solutions. Yet, these nodal characteristics were nonlinearly related to the solutions. Given the example networks, it should be noted that distance to the focal node was consistently related to the quality of the solution as this lowers connectivity length costs, while centrality was intermittently correlated with solution quality it provides greater benefit through more connections. Aside from the distance characteristic, these nodal characteristics also varied in their correlation (sign and magnitude) with solution quality for different types of networks. This makes it difficult to exclude or prioritize specific nodal characteristics for local network connectivity optimization heuristics. This could arise from the four following issues: (a) the curse of dimensionality, i.e., large sparse subspaces in the solution space; (b) the nodal characteristics are highly correlated with each other; (c) outliers; and (d) the nodal characteristics are heterogeneous across the network. Results from the street networks in the transportation case-study found that the clustering coefficient was a poor measure due to the lack of triplets in the networks.

The optimization heuristics save computational time but vary considerably in their ability to find (near) optimal solutions. The stochastic hill climbing search was not effective due to the large neighborhood search space explored. In our experiment, the number of solutions checked at each iteration is >300 and resulted in a skewed probability distribution of objective values favoring the selection of low values. This degraded the efficiency of the method resulting in the selection of poor solutions. The variable neighborhood search method was similarly not reliable because of the significantly large neighborhood search space (the number of possible solutions explored at a given iteration could be >5,000), and had intermediate results with cost and benefit deviations. The simulated annealing heuristic consistently took longer to converge than the other optimization methods from the exploration of suboptimal solutions prior to moving towards better solutions, yet it was able to converge to values close to the optimal solution.

The computational costs and the variance in the importance of nodal characteristics for the random networks and real-world systems highlights the need for a heuristic that is able to quickly and effectively explore the solution space without getting stuck in a local optimum. The genetic algorithm provided in this work offers a solution to these issues and outperformed the other algorithms in terms of the consistently higher solution precision and accuracy. The genetic algorithm is able to dynamically reduce the size of the neighborhood search space and what variables to analyze. This reduction in the local solution search space allows the genetic algorithm presented here to converge on solutions near the optimal in a timely fashion. The heuristic is also able to compare solutions from distant search spaces with nonzero probability, thereby avoiding local optimums. The experiments indicate the power of biologically inspired algorithms to effectively explore multidimensional spaces (commonly found in natural systems) and their potential use in a wide variety of disciplines, including the specific applications for planning and crisis management presented above. The combinatorial optimization techniques employed here to identify and evaluate new street connections can also complement the optimization approaches used for other transportation planning problems, such as greenway planning (Linehan, Gross & Finn, 1995), bus stop locations (Ibeas et al., 2010; Delmelle, Shuping & Murray, 2012), and health care accessibility (Gu, Wang & McGregor, 2010).

There are several research directions from these proposed methods. Application of these methods and heuristics can be tested on multi-level networks, such as telecommunication systems, higher dimensional real-world networks (transportation networks with elevation), directed networks, and additional planar random networks (e.g., Gabriel graphs) and this should be conducted. Different distance measures to the focal node, such as the Hamming distance, could also be evaluated for different applications, and other real world examples should be used for analysis. The methods presented here simplified the costs and did not account for many real-world barriers that may restrict optimal new connections found through the heuristics. For example, the transportation case study did not include legal considerations, such as right-of-way, or physical barriers, such as highways or rivers. Furthermore, the methods presented here do not evaluate whether the new connections intersect existing edges, as in existing transportation networks, and attempts to incorporate such a feature resulted in unrealistic computational times.

Supplemental Information

Supplemental Information 1 Algorithm pseudocode and additional figures

Click here for additional data file.

Supplemental Information 2 Algorithm Code

Matlab code for each algorithm shown in the Supplemental Materials.

Click here for additional data file.

Supplemental Information 3 School adjacency matrices

Matlab files containing the adjacency matrices for the ten schools used in the real world case study.

Click here for additional data file.

We would like to thank Alex Zendel (GIS Analyst at the Knoxville-Knox County Metropolitan Planning Commission) for providing the street networks and residential data around the schools.

Additional Information and Declarations

Competing Interests

Author Contributions

Data Availability

The authors declare there are no competing interests.

Jeremy Auerbach conceived and designed the experiments, performed the experiments, analyzed the data, performed the computation work, prepared figures and/or tables, authored or reviewed drafts of the paper, and approved the final draft.

Hyun Kim conceived and designed the experiments, prepared figures and/or tables, authored or reviewed drafts of the paper, and approved the final draft.

The following information was supplied regarding data availability:

The pseudocode for the algorithms, algorithm codes and the school network adjacency matrices are available in Supplemental Information. MAT files can be imported into the open source software R.

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
