# Peer review of "Local network connectivity optimization: an evaluation of heuristics applied to complex spatial networks, a transportation case study, and a spatial social network"

_PeerJ Computer Science, doi:10.7717/peerj-cs.605_

## Round 0.1 · original submission · Major Revisions

I'd appreciate it if in the revised version you can address the issues raised by all the reviewers.

Reviewer 1 ·

Basic reporting

a) The language of this article is professional, clear and technical.
b) The introduction and background part have elaborated the gap and problem this work aiming to solve. Related literatures or prior works have been reviewed in introduction part.
c) The structure of this article is identical to the requirement of acceptable format. Figures in this article also support the corresponding content. However, in my option, all figures seem to be blurred and authors should try to solve this problem.
d) All the results in this article and supplementary materials can support the hypotheses or problem definition in this article.
e) Symbol definitions are clear and relevant.

Experimental design

a) This article is in accordance with the Aims and Scope of the journal.
b) Question definition (optimizing network connectivity around a specific node with the introduction of new edges) and gap between prior works and this article (Optimizing the network connectivity of additional edges attempts to maximize the number of nodes within a given distance to a focal node to be connected and minimizing the number and length of additional connections) are clear and the following content also discuss the pathway to fill this gap.
c) The experimental design basically conforms the ethical standard. A study of urban transportation planning in experimental part shows the effectiveness of proposed solution path even though the other results are obtained by performing algorithm on random networks. I think this experimental design is acceptable.
d) Method introduction or discussion is clear.

Validity of the findings

a) The novelty of this article is acceptable. It really solves the problem mentioned in introduction and this problem is actually meaningful in the real world. Results data and discussion are sufficient.
b) The conclusion part is well stated and summarize the whole content in the article.

Additional comments

Generally speaking, it is an interesting work. The problem it solves seems to be minor but actually improve the related researches of network connectivity optimization. By reviewing this paper, I think authors devote themselves to solve this problem step by step. However, I have just one question or suggestion for authors: I hope authors can perform their method on one more real-world network rather than random networks. The transportation network is good. Another study on the real-world networks such as wireless networks or vehicular social networks is encouraged.

Reviewer 2 ·

Basic reporting

no comment

Experimental design

no comment

Validity of the findings

no comment

Additional comments

In "Network connectivity optimization: An evaluation of heuristics applied to complex networks and a transportation case study" authors provide new methods for a problem encountered in systems across many different fields -- namely, optimizing network connectivity to maximize the number of nodes within a given distance to a focal node and then minimizing the number and length of additional connections. This is important in several domains including transportation planning, telecommunications networks, and geospatial analysis.

I have very much enjoyed reading this paper. I find it comprehensive and clearly written, and introducing new, timely, and important results that will surely also inspire future research along these lines. The provided random spatial network formulation can be beneficial to complex systems research, network theory, and interdisciplinary applications. The introduction is also very comprehensive and informative, particularly for the PeerJ Computer Science audience. For these reasons, I am warmly in favor of publication subject only to minor revisions.

In terms of related works and reviews, I would recommend Saving human lives: What complexity science and information systems can contribute, J. Stat. Phys. 158, 735-781 (2015) where such research has been advocated for.

I would also encourage the authors to extend the abstract more with the key results. As it is, the abstract is a little thin and does not quite convey the interesting results that follow in the main paper.

Some references contain errors, missing or incorrect information, and inconsistent formatting. References should thus be corrected with the best care.

Apart from this, I am happy to congratulate the authors to an excellent contribution.

Reviewer 3 ·

Basic reporting

This paper studied the network connectivity optimization problem of minimizing the number of nodes within a given distance to a focal node by adding an edge. The authors mainly used a genetic algorithm to solve the problem. The ideas and results are good, but I think it is not suitable to be published in its current form due to the bad writing. A lot of necessary explanatory information was omitted, which makes readers unable to fully and correctly understand the thoughts the authors want to deliver. For instance, in Equations (2), (3), (4), (5), and (7), what is the meanings of C(i,j), B(i,j), O^t, gene(i, k)? And in the “Local search methodology”, the authors did not give a detail description of the crucial two-level selection process. Besides, other problems exit. For instance, sub-figures (B) and (C) of Figure 2 are vague and unreadable. And in Equation (1), it is confusing that the authors tried to maximize the cost of the newly added edge.

Experimental design

ok

Validity of the findings

ok

Additional comments

This paper studied the network connectivity optimization problem of minimizing the number of nodes within a given distance to a focal node by adding an edge. The authors mainly used a genetic algorithm to solve the problem. The ideas and results are good, but I think it is not suitable to be published in its current form due to the bad writing. A lot of necessary explanatory information was omitted, which makes readers unable to fully and correctly understand the thoughts the authors want to deliver. For instance, in Equations (2), (3), (4), (5), and (7), what is the meanings of C(i,j), B(i,j), O^t, gene(i, k)? And in the “Local search methodology”, the authors did not give a detail description of the crucial two-level selection process. Besides, other problems exit. For instance, sub-figures (B) and (C) of Figure 2 are vague and unreadable. And in Equation (1), it is confusing that the authors tried to maximize the cost of the newly added edge.

---

## Round 0.2 · accepted · Accept

I appreciate the thorough revision the authors have made. All reviewers are happy with the current version for publication.

Reviewer 1 ·

Basic reporting

a) The language of this article is professional, clear, and technical.
b) The introduction and background part have elaborated the gap and problem this work aiming to solve. Related literature or prior works have been reviewed in the introduction part.
c) The structure of this article is identical to the requirement of an acceptable format. Figures in this article also support the corresponding content.
d) All the results in this article and supplementary materials can support the hypotheses or problem definition in this article.
e) Symbol definitions are clear and relevant.

Experimental design

a) This article is in accordance with the Aims and Scope of the journal.
b) Question definition (optimizing network connectivity around a specific node with the introduction of new edges) and the gap between prior works and this article (Optimizing the network connectivity of additional edges attempts to maximize the number of nodes within a given distance to a focal node to be connected and minimizing the number and length of additional connections) is clear and the following content also discuss the pathway to fill this gap.
c) The experimental design basically conforms to the ethical standard.
d) Methods introduction or discussion is clear.

Validity of the findings

a) The novelty of this article is acceptable. It really solves the problem mentioned in the introduction and this problem is actually meaningful in the real world. Results data and discussion are sufficient.
b) The conclusion part is well stated and summarizes the whole content of the article.

Additional comments

Generally speaking, I am very satisfied with this version. The authors have already modified the papers based on the reviewers' comments. More specifically, I suggest the authors add a case study of the transportation networks and I have seen the improvement in this version. The authors give a detailed response to other reviewers' comments. The enhancement of this paper can be observed.

Reviewer 2 ·

Basic reporting

no comment

Experimental design

everything OK

Validity of the findings

everything valid

Additional comments

The authors have revised their manuscript comprehensively and with love to detail. I warmly recommend publication in present form.

Reviewer 3 ·

Basic reporting

no comment

Experimental design

no comment

Validity of the findings

no comment

Additional comments

The authors has addressed all of my previous concerns.